# Teaching Data Science with Literate Programming Tools

**Marcus Birkenkrahe**

Department of Math and Science, Lyon College, Batesville, AR 72501, USA; birkenkrahe@lyon.edu

**Abstract:** This paper presents a case study on using Emacs and Org-mode for literate programming in undergraduate computer and data science courses. Over three academic terms, the author mandated these tools across courses in R, Python, C++, SQL, and more. The onboarding relied on simplified Emacs tutorials and starter configurations. Students gained proficiency after undertaking initial practice. Live coding sessions demonstrated the flexible instruction enabled by literate notebooks. Assignments and projects required documentation alongside functional code. Student feedback showed enthusiasm for learning a versatile IDE, despite some frustration with the learning curve. Skilled students highlighted efficiency gains in a unified environment. However, the uneven adoption of documentation practices pointed to a need for better incorporation into grading. Additionally, some students found Emacs unintuitive, desiring more accessible options. This highlights a need to match tools to skill levels, potentially starting novices with graphical IDEs before introducing Emacs. The key takeaways are as follows: literate programming aids comprehension but requires rigorous onboarding and reinforcement, and Emacs excels for advanced workflows but has a steep initial curve. With proper support, these tools show promise for data science education.

**Keywords:** data science; literate programming; teaching; Emacs; org-mode; IDE; case study

## 1. Introduction

The author began teaching data science at a small liberal arts college in 2021, after a long career of teaching business informatics courses at a German business school. The COVID-19 pandemic prompted him to look for different ways of working with students in the classroom.

A couple of years previously, the author had returned to an old friend from his days as a graduate student of physics, GNU Emacs, which is tersely described as an "extensible, customizable text editor" [1]. For programmers, GNU Emacs is an early tool for "literate programming", which assembles documentation, code, and output in one text document that can be converted to either source code for compilation or a document for printing [2]. For humans, literate programs are easier to understand, debug, and maintain.

In production data science, literate programming is the norm and not the exception, thanks to interactive notebooks that were first popularized by IPython and the Jupyter project [3]. Today, every development platform and IDE offers notebooks such as Google Colaboratory, RStudio by Posit, Kaggle by Google, and dozens of others. This is partly due to the interdisciplinary character of data science, which relies on programming, mathematics, and domain knowledge and skills, and which is used by practitioners with diverse backgrounds who are often not trained in computer science, including biologists, psychologists, sociologists, and medical professionals.

The path to learning data science is highly fragmented. It involves a choice of programming languages (e.g., Python vs. R), data visualization techniques, mathematical and process modeling, and knowledge of computing infrastructure [4].

When teaching data science, a core problem concerns the need to provide enough of an overview of all these aspects to enable students to solve real-world problems. In his classes, the author employs a triadic progression of didactic concepts: instruction, interaction, and

immersion [5]. As a teacher, he had always used the ancient GNU Emacs editor (created in 1975, first launched in 1985, and first used by him in 1991) and the more modern Org-mode package for Emacs, while the students used a variety of different platforms, such as GitHub for lecture materials, Google Colaboratory for coding along, and DataCamp and Canvas for assignments and tests.

Then, at EmacsConf in November 2021, the author watched the educator Daniel German explain how he uses Org-mode to prepare and present teaching materials to his students when teaching programming in a variety of languages [6]. This emboldened the author to embark on an ambitious plan: to mandate Emacs and Org-mode as the central platform for interaction, instruction, and immersion in all his courses.

This was ambitious because Emacs is said to have a steep learning curve and to be useful mainly to a small group of devoted developers and professionals [7]. However, the Internet has experienced something of an Emacs renaissance, with several popular YouTube channels and blogs featuring Emacs and presenting it as a viable, open-source alternative to commercial products like VS Code by Microsoft or RStudio by Posit [8–10]. These authors prefer Emacs because of its openness, its flexibility, and its stability. For data scientists, Emacs' ability, in concert with Org-mode, to effortlessly work with many programming languages in the same notebook is especially attractive.

When the author started using Emacs in class in spring 2022, the students liked it, and so he kept going until the end of the 2023 spring term.

In this paper, the author will present and reflect on the choice of Emacs and Org-mode as a mandatory literate programming tool for teaching data science in a variety of undergraduate courses for the programming languages R, SQL, SQLite, C/C++, bash, databases, data visualization, machine learning, and operating systems over three academic terms at a small college in rural Arkansas.

### 1.1. The Theory and Practice of Teaching Data Science

As a recently developed field of interdisciplinary study and practice, data science is too young to have a well-established "best practice" for teaching [11]. However, there is basic agreement about what students need to learn in each of its disciplines: at the center of most courses and textbooks stands a workflow that begins with data acquisition and ends with storytelling. Between these two stages, data need to be stored, managed, transformed, modeled, analyzed, and visualized [12]. Figure 1 shows a shortened version of this flow with the four essential phases of any data science project—from cleaning the data for analysis, through modeling and visualization, and, finally, to presenting insights. This data science workflow leads to different job titles, such as data engineer, data analyst, or machine learning scientist, each of whom oversees different parts of the pipeline, while data scientists take charge of the whole pipeline.

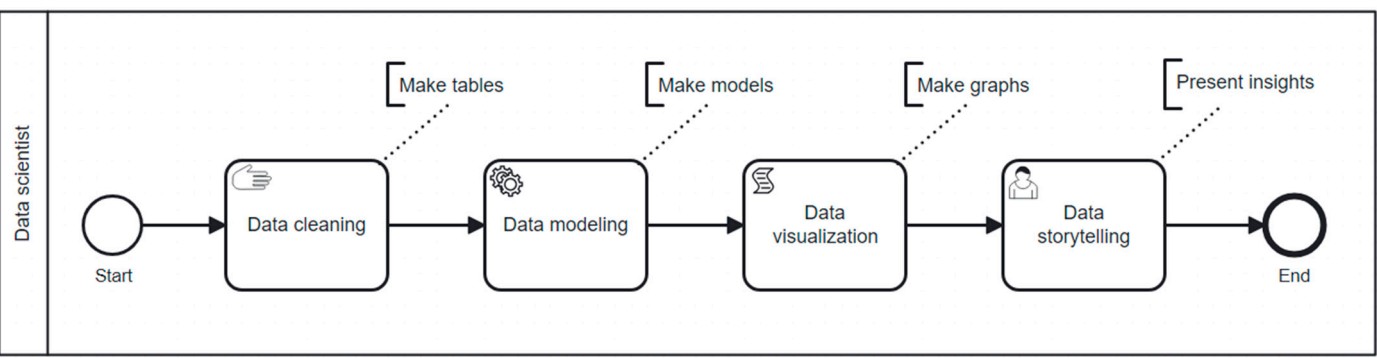

**Figure 1.** Simplified data science workflow.

Until a couple of years ago, data science education focused on graduate-level programs for people with undergraduate degrees in computer science, software engineering, mathematics, and statistics, or on a specific data domains like biology, psychology, or

business [13]. As the demand for data science graduates increased, more and more under-graduate programs sprang up.

Mastering the data science workflow requires many diverse skills. For example, the data science major at Lyon College requires core competence in computer science, mathematics, and statistics, two data science specialization courses (e.g., visualization and machine learning), and two domain-specific specializations (social science/humanities, business/economics, or science). It includes training in R, Python, SQL, C/C++, or Java, plus the foundations of digital logic, database design, and operating systems, along with bash shell scripting and exposure to Git version control software [14].

The teaching response to this ambitious set of skills has been to create infrastructures that integrate different tools so that students can focus on the data science task at hand—e.g., importing and cleaning a dataset, obtaining an overview of the dataset structure, and creating exploratory visualizations.

Training platforms such as DataCamp have perfected this approach, offering smooth sailing through different topics in only a few hours while hiding many of the tricky aspects—like finding and loading suitable software packages, managing files and processes, mastering the interface between graphics and the operating system, and so on. Integrated development environments (IDEs) like RStudio, VS Code, Google Colaboratory, Spyder, and Anaconda have taken a similar path. The same methodological attitude lies behind the recent trend toward "code intelligence", i.e., automatic comment generation and code completion, enabled by large language models [15].

The problem with these integrated infrastructures is that they do not represent the systemic structural messiness of the real world, and they do not train students specifically in meeting real world requirements, especially transparency and reproducibility [16]. This would, for example, include having to struggle with setting up, managing, and debugging a multi-part work environment on one's own and making it work remotely with other developers, who may or may not have the same setup.

Literate programming aligns well with the goals of reproducible, transparent, and open data science education. The literate programming methodology promoted through Emacs and Org-Mode allows students to unify theory, code, and output within a single coherent text document, using only a single application while still being in full control of the work environment and its customizations.

## 1.2. The Rationale for Using Emacs as an IDE and Its Learning Curve

As an IDE, Emacs provides a unique set of advantages, particularly within the realm of data science education [17]. Its broad customizability, extensive programming language support, and open-source nature underscore its utility. However, it is important to acknowledge that Emacs also introduces certain complexities that can make it challenging for newcomers to learn.

Emacs' customizability is a double-edged sword. On the one hand, it empowers users to tailor their programming environment to their specific needs—an ideal feature for a data science course that requires different programming tools and workflows. Yet, the flip side of this flexibility is that Emacs lacks a definitive 'out-of-the-box' setup. Beginners must invest significant effort into learning how to configure Emacs effectively. It is not just about learning a new tool; it is about shaping that tool to fit one's needs.

Additionally, while Emacs' support for a wide range of programming languages adds to its versatility, it also requires the user to have a deeper understanding of these languages and their integration into the Emacs ecosystem. Navigating these integrations can prove challenging for novices, who may need to grapple not just with the intricacies of the languages themselves, but also with how to effectively set up and use Emacs to code in these languages.

Furthermore, while being open source offers advantages in terms of reliability and flexibility, it also introduces another layer of complexity. Users often must sift through an

abundance of community-generated resources, deciphering what is relevant and reliable, which can be daunting and time-consuming for beginners.

Lastly, Emacs' extensibility, through packages such as "Emacs Speaks Statistics" (ESS) [18] that are designed for statistical programming and data analysis, solidifies its standing as a versatile tool for data science tasks. This extensibility, however, comes with a learning curve. The sheer number of available packages can be overwhelming to new users, requiring them to understand the utility and application of each one.

When using Emacs in class as a teacher, and especially when mandating its use by students, it is important to rein in expectations and provide a pre-configured environment. Figure 2 shows the minimal Emacs configuration file provided to the students. It enables them to:

1.  Run code in Emacs in C/C++, R, SQL, SQLite, R, Python, and bash.
2.  Update Emacs packages from a central repository.
3.  Create code blocks easily with skeleton commands.
4.  Auto-load the ESS package for using R in Emacs.
5.  Disable toolbar and graphical menu bars (not being able to use graphical menus discourages use of the mouse and helps users to rely on the keyboard as the only and faster way to get things done).

```
MINGW64:/c/Users/birkenkrahe/Downloads                              —    □    ×
(require 'ob-sqlite)
(require 'ob-sql)
(require 'python)
(require 'ob-emacs-lisp)
(require 'ob-R)
(require 'ob-C)
(require 'ob-shell)
(require 'ob-python)
(org-babel-do-load-languages
 'org-babel-load-languages
 '((R . t)
   (sql . t)
   (python . t)
   (emacs-lisp . t)
   (C . t)))
(setq org-confirm-babel-evaluate nil
      org-src-fontify-natively t
      org-src-tab-acts-natively t)
(require 'org-tempo)
(require 'package)
(add-to-list 'package-archives
             '("melpa-stable" . "https://stable.melpa.org/packages/"))
(global-set-key (kbd "<f6>") 'org-display-inline-images)
(global-set-key (kbd "<f7>") 'org-remove-inline-images)
(add-to-list 'load-path "~/.emacs.d/elpa/ess-20221121.1627")
(load "ess-autoloads")
(tool-bar-mode -1)
(menu-bar-mode -1)
1-DD-\----F1   .emacs        Top (28,19)    (ELisp/d ivy ElDoc) --------
```

**Figure 2.** Emacs configuration file `.emacs`.

### 1.3. The Rationale for Using Org-Mode as a Literate Programming Tool

Org-mode is a structured plain-text format with notebook-like live code execution; it offers an ideal platform for literate programming, a methodology that intermingles code, documentation, and output within a singular document [19]. Conceived by Donald Knuth in 1984 [2], this practice promotes the creation of programs that are not just functional but

are also easy to understand, debug, and maintain. In the context of data science education, this form of programming plays a key role in unifying theory and practice, enabling students to visualize the results of their code in parallel to the theoretical constructs being explained.

For instance, suppose a data science student is working on a machine learning project to predict housing prices. Using Org-mode, they can describe the theoretical concepts of their chosen regression model, then input the corresponding code, and, finally, display the generated outputs, all in one unified document. This seamless presentation not only makes it easier for the student to comprehend the link between theory and implementation, but it also enhances the readability for others who might review or collaborate on the project.

Org-mode also bolsters the creation of reproducible research documents, a cornerstone of modern data science. Integrating code, results, and narrative text, Org-mode documents are ideal for assignments, projects, and collaborative work. For example, a student could use Org-mode to write a report on a data cleaning project. The report could contain blocks of R code for handling missing data, interspersed with explanations of why certain strategies were chosen. The output from the code (such as summary statistics or visualizations) can be included directly below the code blocks, creating a comprehensive, easily understandable narrative—all of this with maximum portability in the text format, not being limited to any reader application or file format.

Furthermore, Org-mode's ability to export these documents to various formats, including HTML, PDF, and LaTeX, eases the process of sharing work. A research team could collaborate on a data analysis project in Org-mode and then export the project as a PDF to share with their client or as an HTML page to publish on the web. The LaTeX export option allows for the creation of formal academic articles, replete with features such as bibliographies and index creation. The opportunities to export into a variety of formats go significantly beyond the abilities of most interactive notebook environments.

In summary, the purpose of literate programming is to allow humans to create a story with rich metadata while coding. The resulting literate programming file can be 'tangled' into source code for compilation, and 'woven' into a document that includes text, code, and output in a publication-ready format.

## 2. Methodology

This study is a design-based research (DBR) study, which seeks to improve educational practices by iteratively designing, implementing, and evaluating interventions [20]. The goal of this study is to develop evidence-based design principles for the use of literate programming tools in a series of linked courses.

The study was conducted in a real higher education setting, and the author of the paper was involved in all aspects of the study, from instigation and design to implementation and evaluation. The author was a participant–observer in the study, which means that he was both a participant in the courses and an observer of the students' learning process.

The evidence for the study was gathered through insights rather than statistical significance. This means that the author looked for patterns and trends in the students' learning, rather than trying to prove a specific hypothesis. The insights were gathered using systemic action research methods, a type of qualitative research that focuses on understanding the complex interactions between people and their environment [21].

Here are some of the key features of the DBR methodology used in this study:

- The study is iterative, meaning that the design of the intervention was refined based on the evaluation of the findings.
- The study is situated in a real-world setting, which allows the findings to be more generalizable.
- The study is participatory, meaning that the author was involved in all aspects of the study.
- The study is qualitative, meaning that the evidence was gathered through insights rather than statistical significance.

For the limitations of the implementation of this methodological program, see below.

## 3. Case Study: Teaching Data Science with Emacs and Org-Mode

This case study details the use of Emacs and Org-mode as central teaching tools across multiple undergraduate data science courses taught by the author. The following sections describe the specific courses involved, the student participants, how instruction was tailored to address knowledge gaps, the Emacs packages and configurations used, onboarding students to these platforms, in-class teaching methods leveraging literate programming, assignment and project implementation, participant feedback and evaluations, and key challenges encountered along with lessons learned. In this multi-term case study spanning a variety of classes and languages, the goal was to assess the potential benefits and limitations of mandating literate programming with Emacs/Org-mode in data science education.

### 3.1. Courses and Participant Profiles

Over the course of three academic terms (spring 2022, fall 2022, and spring 2023), the author used Emacs and Org-mode as the primary tools for teaching a variety of undergraduate courses. The class sizes ranged between 6 and 28 participants. The participants included undergraduate students from all levels, including freshman, sophomore, junior, and senior. The students were from different majors, including computer science, mathematics, and engineering. None of the participants had had any previous contact with Emacs or Org-mode.

Table 1 shows all the courses involved in this case study along with the programming language mainly used in the course, the course level (1–4) and denomination (CSC = computer science course, DSC = data science course, MTH = mathematics course), and the number of participants in each course.

**Table 1.** Courses involved in the case study (CSC: computer science, DSC: data science).

| Course Name (Main Language) | Level | When | Participants |
|---|---|---|---|
| Intro to programming in C++ (C/C++) | CSC 100 | Spring 22/23 Summer 22 | 13/13 6 |
| Intro to data science (R) | DSC 105 | Fall 22 | 13 |
| Intro to advanced data science (R) | DSC 205 | Spring 23 | 13 |
| Digital humanities—text mining (R) | CSC 105 | Spring 23 | 6 |
| Database theory and applications (SQLite) | CSC 330 | Spring 22 | 28 |
| Data visualization (R) | DSC 302 | Fall 22 | 15 |
| Machine learning (R) | DSC 305 | Spring 23 | 20 |
| Operating systems (bash) | CSC 420 | Spring 22 | 22 |
| Applied math in data science (R) | DSC 482/MTH 445 | Fall 22 | 20 |

The material for all these courses is freely available in the author's public GitHub repositories (distributed under a GNU General Public License v3.0).

### 3.2. Addressing Specific Student Knowledge Gaps

In several courses, the students had to fill out an entry survey before the first session to help the author establish a baseline of what they already knew. The survey results revealed that most students did not know how to open a command line terminal, and that most students knew at most one programming language. These findings helped the instructor to tailor instructions to the specific needs of the students and help them develop a better understanding of their operating system and the computing infrastructure.

Specifically, the author addressed the following student needs to improve and enable basic computing infrastructure knowledge:

- Opening and using the command line interface (CLI). A tutorial was created that showed students how to open the command line terminal and how to navigate the

file system from the command line. Participants were also provided with exercises to practice these skills.

- Exploring the file system from the command line. The students were shown how to use the command line to navigate the file system, including how to create, delete, and rename files and directories. They were also shown how to use the command line to search for files and directories.
- Explaining and practicing CLI compilation or, equivalently, running scripts using an interpreter. The difference between compilation and interpretation was explained and students were shown how to compile and run programs from the command line. They were provided with exercises to practice these skills.
- Using the shell and creating shell scripts in Emacs. Students were shown how to use the shell to interact with the operating system and how to create small shell scripts. They were also shown how to use Emacs to edit shell scripts.
- Options for editing, executing, and debugging programs. For instance, when it came to the statistical programming language R, the students were introduced to several alternatives. These alternatives included: running R through a console in a terminal; utilizing the GUI provided by base R; accessing R online through platforms like Google Colaboratory and Replit; utilizing the DataCamp workspace with Jupyter Lab; using the RStudio IDE: and integrating R within Emacs, including running it in the background while executing an Org-mode code block.

### 3.3. Emacs Version and Packages Used

The author's own Emacs configuration did not require a deep understanding of Emacs or Emacs-Lisp (the language used to configure and program the Emacs editor).

He used the vanilla GNU Emacs editor (version 28.2 at the time of writing) for Windows 10. Because it is easy to customize Emacs, in line with the general GNU philosophy, it is possible to adapt the Emacs to pretty much any workflow, aesthetic preference, or keyboard and language settings [1].

In those courses where R was the primary programming language, the author used Org-mode in conjunction with the ESS package. For version control with Git, he used the magit package, integrated with GitHub [22].

For courses teaching C/C++, SQLite, or bash, Org-mode with the built-in Babel extension for multi-lingual code was sufficient to run code [23]. Unlike all other interactive notebook environments, Emacs allows code blocks of different languages in one and the same file.

For classroom presentations, the author used the Emacs org-present package to render Org-mode files in the presentation format, and the modus-themes package as the general Emacs theme. To make the code clearer, he used the rainbow-delimiters package, and org-appear to hide the emphasis markers used, e.g., to highlight code.

Figure 3 shows an Emacs Org-mode buffer with a "Hello World" code block and corresponding output in (from the top) five programming languages: Python, SQLite, C, R, and bash. This works only, of course, if the respective languages are installed and if the environment is initialized properly so that Emacs can find the interpreters or compilers needed to execute the code.

```
#+begin_src python :results output
  print('Hello world')
#+end_src

#+RESULTS:
: Hello world

#+begin_src sqlite :db test.db :results output
  SELECT 'Hello world'
#+end_src

#+RESULTS:
: "Hello world"

#+begin_src C :results output :main yes :includes <stdio.h>
  printf("Hello world");
#+end_src

#+RESULTS:
: Hello world

#+begin_src R :results output
  print('Hello world')
#+end_src

#+RESULTS:
: [1] "Hello world"

#+begin_src sh :results output
  echo "Hello world"
#+end_src

#+RESULTS:
: Hello world
```

```
1 -\**-   babel.org      All (30,20)    (Org org-ai ivy)
Code block evaluation complete (took 0.2s).
```

**Figure 3.** Emacs Org-mode buffer with the "hello world" program in five languages.

*3.4. Tools Used across All Courses*

When the author taught data science at a German business school, he used twelve different tools in the classroom [24]. When he began to use Emacs for preparation, lectures, and practice, he could reduce the number of tools drastically to four tools plus Emacs:

GitHub. GitHub was the central repository for all course materials except quizzes, tests, and grade data. Git was fully integrated with Emacs, allowing the author to work in different locations while maintaining a central, up-to-date material collection for the students. Students received minimal instruction in using GitHub (registration was not mandatory).

DataCamp. The author used the DataCamp online classroom in all his courses for home assignments: students were required to complete lessons from the relevant DataCamp courses (e.g., "Introduction to R", or "Understanding machine learning") on a regular basis.

Learning Management System. Participant activities in the LMS included weekly online quizzes and tests and home programming assignments, the solutions to which were submitted in the LMS. The students could follow their personal progress at any time using

the built-in, up-to-date gradebook. Both Schoology (Spring 2022) and Canvas (Fall 2022, Spring 2023) were used in this way.

Zoom. Zoom was used to share the author's screen with the students so that they could discern details, e.g., when coding along, and to record each session for later viewing.

DataCamp, the LMS, and Zoom remained decoupled from the Emacs-oriented workflow.

### 3.5. Teaching Emacs and Org-Mode Onboarding

A key part of the study was teaching students how to use Emacs and Org-mode, since most were unfamiliar with these tools. To facilitate onboarding, the author developed a simplified Emacs tutorial focused on the basics alongside a brief cheat sheet of the most common commands, including:

- Navigation and modes;
- Managing files and buffers;
- Customizing the interface;
- Keyboard shortcuts.

This hands-on tutorial was delivered interactively during class. Students followed along on their computers as the author demonstrated Emacs features; this was followed by practice in applying the skills through simple editing exercises.

Additionally, the author provided sample initialization and configuration files for Emacs and Org-mode. Students could use these as a starting point to tailor the tools to their needs. Throughout the term, troubleshooting help was available during classes and office hours.

No separate Org-mode tutorial was used. Instead, Org-mode skills, such as adding metadata and using code blocks, were taught through integrated examples during regular in-class programming exercises.

Though the students were not necessarily conscious of markup methods, most had encountered the effects of separating content and layout instructions, e.g., in an HTML file. Hence, the way that Org-mode metadata were used to control output and appearance was not entirely foreign to them.

This combination of hands-on practice, custom configurations, and integrated learning was used to help participants quickly gain proficiency. By the second week of classes, most students were able to use Emacs and Org-mode competently for their assignments. Frequent reinforcement of skills also contributed to students' learning.

Figure 4 shows the top of the tutorial (as a Markdown file on GitHub) with the table of contents.

### 3.6. In-Class Instruction

The course participants were instructed directly in Emacs using Org-mode files only. The author's Emacs screen, showing an Org-mode file (aka, a notebook), was shared with the students (and recorded) via Zoom.

The instructor would demonstrate practical skills and the students would use their own Emacs Org-mode files to either repeat them or solve simple exercises in class. Depending on the complexity of the material, the author would prepare practice files or ask the students to create their own Org-mode files from scratch.

Completed Org-mode files from class coding sessions could be uploaded to the LMS for credit and would regularly be checked and commented upon in the students' notebooks.

Delivering live code presentations with Org-mode provided flexibility: the author could evaluate code blocks on the fly, modify examples, and expand explanations using the literate programming approach. This interactivity would have been impossible to achieve using static, slide-based programs, and is harder to accomplish in an IDE where code and documentation (apart from program comments) are not in the same location.

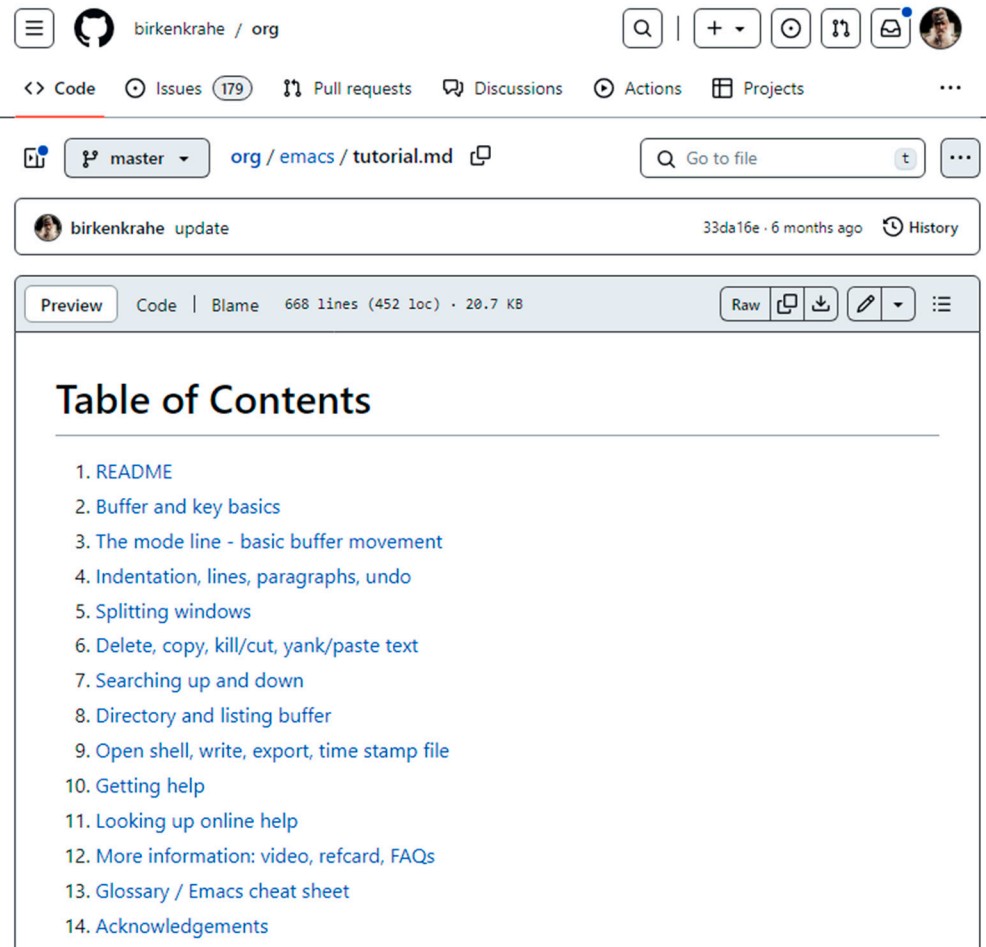

**Figure 4.** Table of contents of the Emacs tutorial on GitHub.

Figure 5 shows the top of an Emacs Org-mode practice file for DSC 205 ("Introduction to advanced data science") used by the students in class in parallel with a lecture on user-defined functions in R. An R source code block was prepared where the students could put their solution. The file header includes file metadata (`#+title`, `#+author`, `#+subtitle`), layout metadata (`#+startup`, `#+options`), and code block metadata—header arguments for an R code block with output to the screen, the code of which runs in an R session buffer named `*R*`. The ":`noweb yes`" argument means that noweb-style chunk substitution is enabled [25].

### 3.7. Assessment

All courses were graded based on different, equally weighted categories, shown in Table 2 as published in the syllabus:

- Multiple-choice tests (of no fewer than 10 questions per week) were administered weekly, covering the week's topics and DataCamp assignments. The test results were reviewed at the start of every week following the test.
- The final exam was a representative selection of those questions from the tests in which the participants had performed worst.
- For home assignments, in-class assignments, and projects, see below.

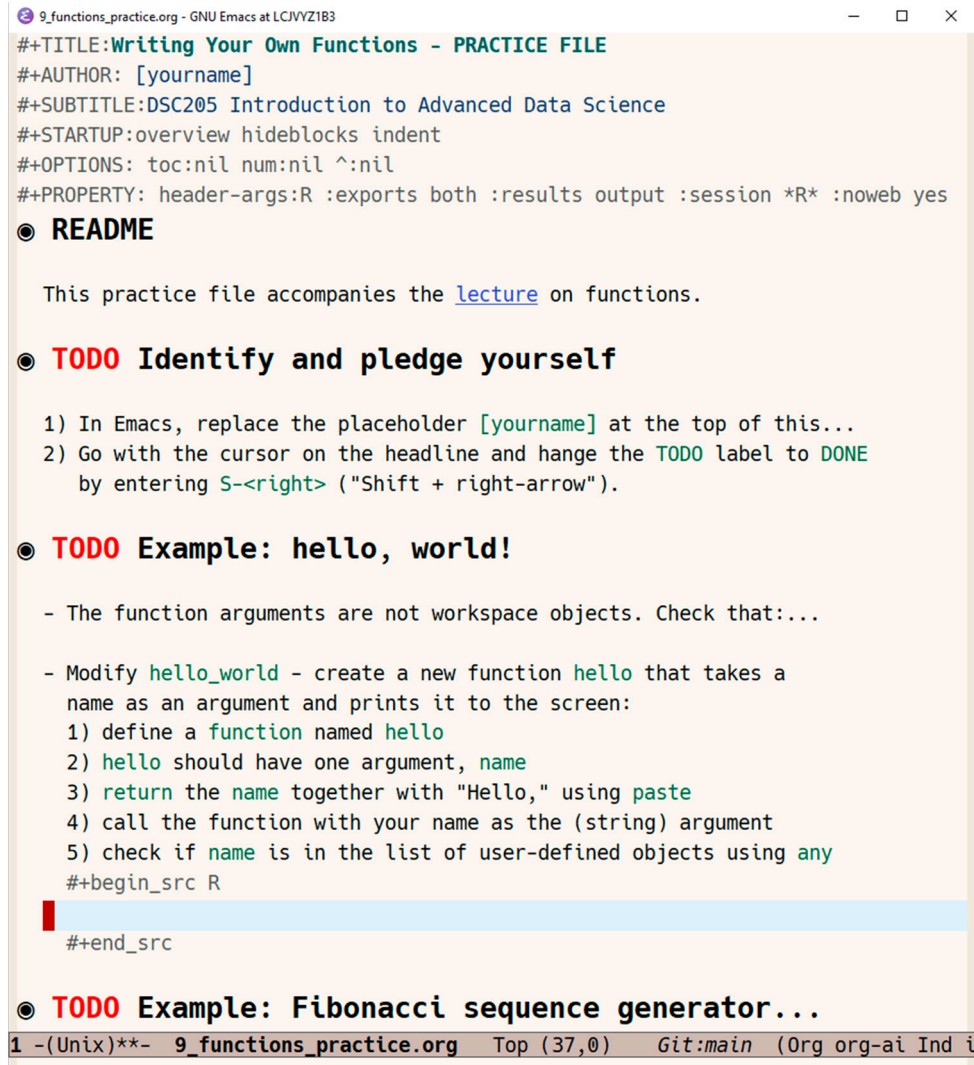

**Figure 5.** Student practice Org-mode file for classroom use (from course DSC 205).

**Table 2.** Grading system published in the syllabus document.

| Requirement | Units | Points per Unit | Total | % of Total |
|---|---|---|---|---|
| Final exam | 1 | 100 | 100 | 20 |
| Home assignments | 10 | 10 | 100 | 20 |
| Class assignments | 10 | 10 | 100 | 20 |
| Project sprint reviews | 5 | 20 | 100 | 20 |
| Multiple-choice tests | 10 | 10 | 100 | 20 |
| Total | | | 500 | 100 |

3.7.1. Assignments

There were three types of coding assignments:

1. Assignments consisting of completing a DataCamp lesson.
2. Home coding assignments (Org-mode notebooks).
3. In-class assignments (Org-mode notebooks).

The DataCamp lessons (for R and SQL) used a modified, customized platform consisting of a series of problems often preceded by an instructional video. The platform consisted of text, a code editor, and a console.

For home and in-class assignments, the students used Emacs as their text editor for writing and debugging code in Org-files. The answers to the programming exercises had to be submitted as Org-mode files complete with documentation, code, sample input, sample output, and references.

Figure 6 shows a programming assignment for the course "Introduction to programming in C++", including the creation of a literate program both in Org-mode and in "tangled" form (i.e., as C source code).

1. Write a program that prompts the user to enter a telephone number in the form `(xxx) xxx-xxxx`, and then displays the number in the form `xxx.xxx.xxxx`.

2. Example input/output of the first program, `phone1.c`:

```
Enter phone number [(xxx) xxx-xxxx]: (870) 456-7890
You entered: 870.456.7890
```

3. Write another program that asks for the input format in the form `xxx\xxx\xxxx`, and then displays the number in the form `(xxx)xxx-xxx`.

4. Example input/output of the second program, `phone2.c`:

```
Enter phone number [xxx\xxx\xxxx]: 870\456\7890
You entered: (870) 456-7890
```

5. Submit one Emacs Org-mode file `phone.org` with both programs in it as code blocks that can be **tangled** as `phone1.c` and `phone2.c`, respectively.

6. The header information of your Org-mode file should look like this:

```
#+TITLE: Phone number conversion
#+AUTHOR: [your name]
#+HONOR: pledged
#+PROPERTY: header-args:C :main yes :includes <stdio.h> :results output :tangle y
es
```

7. Tip: some characters, like `\` are protected because they are part of the file `PATH`. If you want to use them, you have to "escape" them with an extra `\`, like the newline character `\n`. So to print (or to scan) the character `\`, you use `\\`.

8. **Here is a short video** ⤷ (9 min) that explains in detail how to get started with this exercise in Emacs + Org-mode + C.

**Figure 6.** Assignment for an introductory course in C (CSC 100).

3.7.2. Projects

In addition to regular assignments, students in almost all courses had to complete a team-based term project using an adapted agile methodology called Scrum, with the author as the product owner [26]. Teams of two to three students could pick a project topic from a provided list or choose an idea of their own idea that was relevant to the course material.

Throughout the term, teams presented their progress during four sprint review meetings, held approximately every three to four weeks. This allowed them to receive ongoing feedback and guidance.

The final project deliverable was a fully documented interactive report in the form of an Emacs Org-mode notebook. At the end of the term, each team also gave a presentation demonstrating their work to the entire class.

The author was able to provide input and assessments during the sprint reviews in addition to evaluating the final report. This helped to simulate a real-world collaborative environment.

In spring 2022, no projects took place, and multiple students provided feedback requesting that they be brought back in future classes. The agile, team-based projects with regular feedback proved engaging and beneficial, based on student responses.

The success of this adapted Scrum approach to team projects in computer science education was described in more detail in a prior publication [27]. The projects help reinforce the practical application of concepts learned in class through hands-on collaborative problem solving.

The use of literate programming and interactive notebooks for the project reports provided unique benefits compared to traditional code-only deliverables. By having to interweave documentation, references, and outputs alongside functional code, students improved their ability to communicate their technical work to an audience in a reproducible format. The requirement to attend to code quality, explanations, and results throughout the project lifecycle, not just at the end, improved the depth of analysis and understanding. Teams had to consciously connect their design decisions and discoveries to academic sources.

Overall, the literate programming approach enhanced student learning and resulted in higher-quality project work, as validated by both the author's assessments and student feedback. The interactive element added unique value, which static reports lack, in demonstrating capabilities and engaging viewers.

Table 3 lists a selection of topics from projects in different courses that were completed and presented by participant teams (for COURSE, see Table 1).

**Table 3.** Selected term project presentation titles.

| PROJECT TOPIC | COURSE | TERM |
|---|---|---|
| Regression models on Twitter impressions | DSC302 | Spring 2023 |
| Reactivity in R Shiny Dashboards | DSC302 | Spring 2023 |
| Exploring data science salaries with ggvis | DSC302 | Spring 2023 |
| Legality of AI art | DSC305 | Spring 2023 |
| ChatGPT in higher education | DSC305 | Spring 2023 |
| Langrangian Polynomial for regression | CSC105 | Fall 2022 |
| An introduction to RcppArmadillo | MTH445 | Fall 2022 |
| Spin rate in Baseball | MTH445 | Fall 2022 |
| Introduction to Power BI | DSC482 | Fall 2022 |

*3.8. Participant Evaluations*

All courses were subject to standardized, online, anonymous participant evaluations. The survey report consisted of general five-point Likert scale questions on course quality and instructor quality, followed by three open questions:

1. What are the best features of this course?
2. Do you have any suggestions for the improvement of this course?
3. Do you have any additional comments on this course and/or the instructor?

The response rate was high, with an average participation of 82.86% and detailed feedback provided on the use of Emacs in class. Tables 4 and 5 contain the answers that relate to the use of Emacs and/or the use of interactive Org-mode notebooks in the first two open questions. The negative answers show that the anonymous evaluation worked in the sense that students were not reluctant to share criticism. The answers also show that the overwhelming number of students who answered were happy with using Emacs and Org-mode as literate programming tools.

**Table 4.** Student answers to "What were the best features of this course?".

| | What Are the Best Features of This Course? |
|---|---|
| 1 | "The use of Emacs. Learning such a powerful tool will, I feel, truly help me excel after college." |
| 2 | "The new possibilities and features that Emacs brings to not only my programming classes but all of them." |
| 3 | "The notebook[s] through emacs were very helpful." |
| 4 | "Emacs and working with .org files" |
| 5 | "Emacs was my favorite part about this course it was a challenging program to use but worth learning." |
| 6 | "The interactive notebooks. While the lecture is going on, being able to use what you are learning at the same time helps to remember the information later on." |
| 7 | "I like how we use Emacs to take notes and code along with the professor, so that we can practice the skills we learn in class." |
| 8 | "The use of emacs and the terminal over a GUI, while difficult at first, is more representative of tools we may use in the future and are very important skills to learn for competent use of computers. The introduction to emacs specifically is quite daunting, but it's useful for those that accept it." |
| 9 | "Being able to be hands on with the programs in class." |
| 10 | "The interactive notebooks that allow students to use what they are learning to use immediately." |
| 11 | "I enjoy the use of emacs as a literate programming environment." |
| 12 | "The constant practices we do during the semester." |
| 13 | "Getting to learn first-hand with pre-planned activities" |
| 14 | "Learning to use Emacs and GitHub." |
| 15 | "The emacs notebooks that allow you to use what you are learning in real time." |
| 17 | "I like the use of emacs because it is a versatile tool that might help us later in our careers." |
| 18 | "I like how interactive the course is." |
| 19 | "Being able to code alongside [the teacher]." |
| 20 | "Learning how to create data and move through my computer without touching my mouse." |
| 21 | "The instructor walks through the coding programs and makes sure you understand. Also stops if you are stuck." |
| 22 | "Using emacs as the text editor also helped me develop coding skills and get more familiar with managing files, switching between buffers, and other skills that will be fundamental in any career in the Computer Science area that I follow after college." |

**Table 5.** Student answers to "Do you have any suggestions for improvement of the course?

| | Do You Have Any Suggestions for Improvement of the Course? |
|---|---|
| 1 | "I really did not like using Emacs. I felt like it complicated things more than it helped, and it took a couple weeks at the beginning of the semester to learn that we could have been learning C or C++." |
| 2 | "It would have been helpful to spend more time on explaining how EMACS works. Even now, I am not confident about my EMACS skills. |
| 3 | "Having a cheat sheet for emacs that focuses on the most common commands used for assignments." |
| 4 | "Emacs is a hassle to work in; since we have so many home assignments in R it would have been very useful to have spent class time downloading it, I missed several home assignments early in the year because I could not figure out how to download it." |
| 5 | "Emacs is a pain to use." |
| 6 | "Try and give the students more challeng[ing] notebooks to complete on their own." |
| 7 | "The installation of Emacs on our PCs was a little stressful on my end." |
| 8 | "eMacs is extremely clunky for me personally, and it crashed at least once every time I tried to work on things outside of class. I'd rather just use a normal compiler at that point. In addition to having to submit the sprint reviews as org-mode files, didn't feel great. I'd rather just submit them as a word document or even a text document. I know several others have also complained about eMacs to me personally." |
| 9 | "Emacs is a convoluted text editor. It would be nice if we used something else that was more streamlined." |

### 3.9. Participant Experiences

In addition to the anonymous evaluations, in Spring 2022, the author asked students to comment on their experience with Emacs in class. The answers were positive throughout. Here, are four exemplary statements (all seniors at the time; student names used with permission):

"It was the first time I had ever used an editing software, and it will definitely have an impact on how I take notes/write code in the future."

(Hunter Perkins)

"I find [Emacs] very helpful. The ability to use it for multiple programming languages alone makes it powerful and worth learning."

(Victor Noppe)

"The ability to have just one app to code in all of these different languages with minimal setup is a breath of fresh air [. . .] a very useful tool despite its learning curve."

(Jacob Wolfrom)

"I learn best with examples and by doing, so when we started doing the Org mode notebooks in class, I really started to learn to program."

(Spencer Rhoden)

## 4. Discussion and Limitations

This section provides a critical examination of the study results, including challenges faced and lessons derived, along with an analysis of the study's limitations. The first subsection details key difficulties encountered in the implementation of Emacs and Org-mode for teaching data science, as well as best practices identified that may inform future pedagogical approaches. The next subsection discusses the role of traditional literate programming in the era of low-code platforms and AI coding assistants, arguing for its continued relevance despite technological shifts. Finally, the limitations of the present case study are discussed, including potential biases, the inherent complexity of the tools assessed, and the restrictions of the evaluation methods employed. While valuable insights emerge, further research is needed to address these limitations and gain a more comprehensive understanding of utilizing literate programming tools in data science education.

### 4.1. Challenges and Lessons Learned

Some key challenges from mandating Emacs as the primary literate programming data science platform, presented in no particular order, are as follows:

- The steep learning curve of Emacs was an initial hurdle for many students. Some skipped the intro tutorial, which slowed their coding progress.
- There was uneven student adoption of documentation practices in Org-mode. Some focused solely on code rather than full literate programming.
- Most students did not utilize references in Org-mode assignments, despite their academic importance.
- A portion of students found Emacs clunky or frustrating to use.
- Technical issues, like Emacs crashing outside of class, caused problems for some students.
- The lessons learned included:
- It is necessary to mandate and follow-up on Emacs tutorial completion early on.
- Documentation skills must be incorporated into grading rubrics.
- The instructor should model best practices in class materials and demos.
- Exemplary student work should be highlighted to showcase the capabilities of the tools.
- Tools should be matched to student coding skill levels. Emacs may ultimately be better-suited to advanced courses.
- Ongoing reinforcement is needed to overcome reluctance and ensure adoption.
- Overall, while powerful, Emacs and Org-mode had a steep initial learning curve. Adoption was mixed, though skilled students demonstrated the tools' potential.

Despite the challenges, using Emacs and Org-mode also had notable benefits for both teaching and learning. The students who took the time to master the tools were able to work efficiently across multiple programming languages within a single environment. Emacs provided a unified framework for coding in languages like Python, R, C++, and more.

Additionally, the literate programming approach enabled by Org-mode increased student comprehension. Integrating code, results, and documentation into one document provided an interactive learning experience that reinforced key concepts.

From the instructor's perspective, Emacs and Org-mode facilitated the flexible delivery of materials. Live coding demonstrations could be performed seamlessly, with the ability to evaluate and modify code blocks spontaneously during lectures.

Overall, students who embraced Emacs and Org-mode highlighted the tools' versatility. They appreciated learning how to navigate an environment common in industry. The instructor also benefited from streamlined lesson planning and presentation. With proper support, these tools demonstrate strong potential for enhancing data science education.

### 4.2. Literate Programming in the Age of Low Code and AI Assistants

The advent of low-code platforms and AI-powered coding assistants has transformed the programming landscape [28,29]. Tools like GitHub Copilot and services such as Bubble.io remove much complexity, allowing those with little traditional coding knowledge to build applications through simple drag-and-drop interfaces and plain-language commands.

Considering this shift towards the simplification and automation of coding, what role remains for traditional literate programming using tools like Emacs and Org-mode? At first glance, these may appear antiquated compared to cutting-edge AI technologies. However, literate programming still provides unique benefits that warrant its continued teaching: most low-code platforms and AI assistants take a functional approach, focused narrowly on generating executable code. Documentation is secondary. This limits developers' ability to understand and explain the inner workings of their programs.

In contrast, literate programming interweaves human-readable documentation with code in one cohesive document. This comprehensive encapsulation of the process makes programs easier to comprehend, debug, maintain, and share.

The hands-on nature of literate programming also builds deeper developer skills. Working directly in environments like Emacs reinforces the user's firm grasp of coding techniques and problem solving. AI assistants can expedite work, but relying solely on their suggestions deprives developers of learning opportunities.

Lastly, literate programming remains essential for many scientific and academic contexts where reproducibility and transparency are paramount. Computational essays and other reproducible documents require human-authored narrative combined with code.

Rather than displacing traditional coding knowledge, low-code environments and AI can complement the skills gained through literate programming. Just as calculator use strengthens rather than atrophies math skills, leveraging automation can free developers to focus on higher-value tasks informed by a deeper understanding of code.

In data science education, foundations in literate programming continue to provide lasting benefits. Emacs and Org-Mode teach vital concepts that are applicable across multiple programming languages and paradigms. This establishes a basis for effectively utilizing AI coding tools.

The landscape has changed, but literate programming's unique merits persist. It remains an essential component of a well-rounded data science education, laying the groundwork for both automation-assisted workflows and reproducible research. The approaches synergize: literate programming supplies understanding, while AI accelerates its application.

### 4.3. Limitations of the Case Study

While this study provides valuable insights into the use of literate programming tools in teaching data science, it is not without its limitations. One of the primary limitations is the potential bias in the sample selection. The study was conducted in a specific educational setting, and the participants were students who had chosen to study data science. Their motivation and interest in the subject might be higher than average, which could have

influenced the results. Therefore, the findings might not be generalizable to all students or educational settings.

Another limitation concerns the inherent complexity and steep learning curve associated with literate programming tools like Emacs and Org-mode. While these tools offer numerous benefits, mastering them requires a significant investment of time and effort. This could potentially deter students who are new to programming or those who are looking for quicker, more straightforward solutions. The study did not fully explore the potential impact of this learning curve on student motivation and engagement.

Furthermore, the study focused primarily on the benefits of using literate programming tools in teaching data science, without fully exploring the potential drawbacks or challenges. For instance, issues related to software compatibility, technical glitches, or difficulties in understanding the syntax of these tools were not thoroughly examined. These factors could potentially diminish the effectiveness of using literate programming tools in a teaching environment.

Lastly, the study did not consider the impact of other pedagogical strategies or teaching aids that could be used in conjunction with literate programming tools. The use of these tools does not exist in a vacuum, and their effectiveness could be influenced by a variety of other factors, such as the quality of instruction, the use of supplementary materials, or the level of student–teacher interaction.

In conclusion, while this study provides a starting point for understanding the potential of using literate programming tools in teaching data science, further research is needed to address these limitations and provide a more comprehensive picture of this complex issue.

## 5. Conclusions and Outlook

The attempt to use Emacs and Org-mode to teach various data science courses yielded mixed results. There were certainly benefits in terms of the versatility and power of these tools for skilled students. However, the initial learning curve proved prohibitive for some students.

The key advantage of this approach was the immersive coding environment enabled by Emacs and Org-mode. Students who embraced the tools were able to work efficiently across multiple languages like Python, R, and C++ within a single IDE. The literate programming methodology also enhanced their understanding by integrating code, results, and documentation.

However, this experience highlighted the fact that Emacs and Org-mode may not be the optimal choice for all data science students, especially those new to programming. Despite the instructor's best efforts at onboarding through tutorials and examples, some students struggled to become proficient. This hampered their learning during practical exercises.

Additionally, not all students adopted the discipline of documentation through Org-mode despite its importance for reproducible analysis. The motivation to fully utilize these tools was uneven.

Moving forward, the author plans to transition his teaching to tools that offer greater accessibility. For R and Python in particular, Jupyter Notebooks and its freely available online implementations, such as Google Colaboratory, seem better suited for getting novice students started with coding. These platforms remove friction while still supporting literacy programming.

For lower-level languages like C++, beginner-friendly, cross-platform IDEs may provide a gentler onramp. The graphical user interface is less intimidating (but also delivers fewer insights).

While Emacs and Org-mode remain excellent options for advanced data science work, they may be better suited as secondary tools introduced after students have built up their core coding skills. The author still plans to cover these tools to some extent, given their prevalence in the field. However, they will no longer be mandatory for all coursework.

This evolution in his teaching reflects lessons learned about matching tools to students' skill levels.

Finding the optimal set of platforms is an ongoing process as new technologies emerge. However, the lessons from this experiment will guide future decisions, with the goal of choosing the tools best suited to each course and level. These priorities are supporting student learning and building practical data science skills.

**Funding:** This research received no external funding.

**Data Availability Statement:** Data available in a publicly accessible repositoryThe data presented in this case study are openly available in GitHub at https://github.com/birkenkrahe (accessed on 31 August 2023).

**Acknowledgments:** The author would like to thank the students at Lyon College who participated in his courses and provided valuable feedback on the use of Emacs and Org-mode. Their willingness to learn these new tools was essential to this study. He also wishes to express his gratitude to the faculty and staff at Lyon College for their support during his visiting professorship from 2021–2023. The opportunity to teach a variety of data science courses and experiment with innovative approaches was invaluable. Finally, the author is grateful to the Berlin School of Economics and Law for granting him academic leave from 2021 to 2023 to pursue this visiting position. The time spent teaching and researching in the United States has expanded his perspectives and capabilities as an educator.

**Conflicts of Interest:** The author declares no conflict of interest.

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
