# Peer review of "Teaching Data Science with Literate Programming Tools"

_digital, doi:10.3390/digital3030015_

Round 1

Reviewer 1 Report

Paper title:  Teaching Data Science with Literate Programming Tools

Specific comments:

1-      What is the main question addressed by the research? The motivation for the study should be further emphasized, particularly; the main advantages of the results in the paper comparing with others should be clearly demonstrated. 

2-      Do you consider the topic original or relevant in the field? Does it
address a specific gap in the field? The research gap is also missing.

3-      What does it add to the subject area compared with other published
material? The comparison with the existing studies is not sufficient.

4-      What specific improvements should the authors consider regarding the
methodology? What further controls should be considered? The modification is not explicitly explained point by point. The mathematical proof to explain the improvement that has been noticed in the proposed method is also missing and mandatory.

5-       Are the conclusions consistent with the evidence and arguments presented
and do they address the main question posed? The abstract is not cleanly written as well as the conclusion. Analysis of the results is missing in the paper. There is a big gap between the results and conclusion.

6-       Are the references appropriate? More references on recent work should be added as the number of references is low (maybe you could include more references from recent years:

7-       Please include any additional comments on the tables and figures. The figures are not clear some of them are so big relevant to the size of the text.

There are some points need to be further clarified:

A.    The motivation for the study should be further emphasized, particularly; the main advantages of the results in the paper comparing with others should be clearly demonstrated. 

B.    The example section needs to be further expanded and include some remarks to show the effectiveness and efficiency of the proposed method, compared with others. 

C.    Some remarks on the main results would be necessary and helpful. 

D.    The limitations of the studies work should be added. 

E.     The research gap also is missing.

Minor editing of English language required

Author Response

Dear reviewer - Thank you for your careful, detailed reading of the paper. I have tried to address your individual points of critique below.

1- What is the main question addressed by the research? The motivation or the study should be further emphasized, particularly; the main advantages of the results in the paper comparing with others should be clearly demonstrated."

The paper is a case study - the purpose is an investigation into using "Emacs and Org-mode as a mandatory literate programming tool for teaching data science" (from the end of the introductory section). Section 1.1 and 1.2 provide the rationale for the case study.

2- Do you consider the topic original or relevant in the field? Does it address a specific gap in the field? The research gap is also missing.

There is no comparable case study as far as I know so I would say it is an original contribution. The introduction attempts to position the case study in current practice.

3- What does it add to the subject area compared with other published material? The comparison with the existing studies is not sufficient.

Section 1.1 establishes the role of the research topic in the field of data science tesaching.

4- What specific improvements should the authors consider regarding the methodology? What further controls should be considered? The modification is not explicitly explained point by point. The mathematical proof to explain the improvement that has been noticed in the proposed method is also missing and mandatory.

I could not think of specific improvements regarding the methodology. This is not a controlled experiment hence I have not emphasised the issue of controls. I do not know what the reviewer means by "the mathematical proof to explain the improvement". I only cite two sources for the methodology - I have used these before for similar (published) didactic case studies.

5- Are the conclusions consistent with the evidence and arguments presented and do they address the main question posed? The abstract is not cleanly written as well as the conclusion. Analysis of the results is missing in the paper. There is a big gap between the results and conclusion.

The conclusions merely summarize the experiences made during the case study and they directly address the question if the described teaching methods were appropriate or not. The discussion (4) provides an analysis in the form of a list of challenges and lessons learnt, and a thorough description of the many limitations of this study.

6- Are the references appropriate? More references on recent work should be added as the number of references is low (maybe you could include more references from recent years:

There are no more recent references specifically for the topic under consideration. There was a renewed interest in literate programming about five years ago (but not in undergraduate education). Most directly relevant references are from 2021-2023: ref. [6-10,13,15-17,22,24,28-29].

7- Please include any additional comments on the tables and figures. The figures are not clear some of them are so big relevant to the size of the text.

The figures and tables are commented in the text leading up to each of them. The size was chosen to make them readable - one figure whose text was not readable, was dropped.

A. The motivation for the study should be further emphasized, particularly; the main advantages of the results in the paper comparing with others should be clearly demonstrated.

See above.

B. The example section needs to be further expanded and include some remarks to show the effectiveness and efficiency of the proposed method, compared with others.

I have no further data than the data provided in the results section. I tried to balance depth of detail and storytelling in the description.

C. Some remarks on the main results would be necessary and helpful.

See above

D. The limitations of the studies work should be added.

See section 4.3.

E. The research gap also is missing.

See above.

Reviewer 2 Report

1.   A theoretically well-founded work with adequate references to the problem being addressed, which, with a series of adjustments, can be a scientifically interesting article.

2.   A literature review section should be given to discuss more recent methods that have been published in international refereed journals. The authors should discuss the advantages and limitations of these methods.

3. The experimentation, which is the fundamental part of their manuscript, should be better described with regard to literature experiences.

4.Results are not explained in detail. The figures used are not readable. Any details about how the results are interpreted would be helpful.

Author Response

Dear reviewer - Thank you for your careful, detailed reading of the paper. I have tried to address your individual points of critique below.

  1. A theoretically well-founded work with adequate references to the problem being addressed, which, with a series of adjustments, can be a scientifically interesting article.

Thank you.

  1. A literature review section should be given to discuss more recent methods that have been published in international refereed journals. The authors should discuss the advantages and limitations of these methods.

I have summarized the state of the field in the introduction. The existing relevant literature is discussed in three sections in general, with respect to data science teaching (1.1), and with respect to the tools used (1.2-1.3).

  1. The experimentation, which is the fundamental part of their manuscript, should be better described with regard to literature experiences.

There is no experimentation, but only a case study, with some of the typical and many other limitations (laid out in section 4.3). The introduction provides the embedding of the case in teaching and tools.

4.Results are not explained in detail. The figures used are not readable. Any details about how the results are interpreted would be helpful.

The results are described in section 3 including subsections on all areas in which data were available. One figure that was not readable was removed (it did not add much to the content). The other figures only contain text in the same (or larger) font as the paper itself. Also, for online publication, figures can be enlarged on an individual basis.

Reviewer 3 Report

Comments to Authors: The manuscript tells a nice story about a fulfilling teaching experience. It provides useful insight into the rationale (and actual methods employed) behind concrete data science courses. However, the manuscript was submitted as an article (MDPI defines this type as "...original research manuscripts. The work should report scientifically sound experiments and provide a substantial amount of new information") and, in my view, the content does not suit this type of submission. Moreover, the intended message is difficult to follow due to the high level of redundancy – putting some of the content in additional/supplementary files might help increase readability.   Major issues: (A) The organization is not consistent enough: if the manuscript is intended as a case study report, then the methodology should be about the case (even part of information in present introduction would belong to the case); if the manuscript is about teaching methods, then a reference or control group is missing. (B) Present section "2.Methodology" is confusing: (b1) it is not clear to what research it refers; (b2) it needs references to the employed methods, namely the design-based research, and systemic action research methods. (C) Results provided in Table 4 and Table 5 (i.e., students' feed-back) should undergo a qualitative analysis (https://en.wikipedia.org/wiki/Qualitative_research) instead of the [numbered] raw sentences' presentation. Moreover, some quantitative feed-back might be of interest. (D) Part of discussions would better fit into the section of results.   Additional comments: (a) I would kindly recommend a thorough language revision – scientific writing follows distinct rules and style, with little similarity to oral communication or colloquial writing. (b) Numbering of level 2 headings (style "heading2") in section "1.Introduction" should be double-checked. (c) First and third person are confusingly interchanged in the introduction. (d) Use of italics should be consistent throughout the manuscript.

(e) Format of the references should be revised.

I would kindly recommend a thorough language revision – scientific writing follows distinct rules and style.

Author Response

Comments to Authors: The manuscript tells a nice story about a fulfilling teaching experience. It provides useful insight into the rationale (and actual methods employed) behind concrete data science courses. However, the manuscript was submitted as an article (MDPI defines this type as "…original research manuscripts. The work should report scientifically sound experiments and provide a substantial amount of new information") and, in my view, the content does not suit this type of submission. Moreover, the intended message is difficult to follow due to the high level of redundancy – putting some of the content in additional/supplementary files might help increase readability.

I am leaving it to MDPI to decide which type of publication is suitable for this case study.

Major issues: (A) The organization is not consistent enough: if the manuscript is intended as a case study report, then the methodology should be about the case (even part of information in present introduction would belong to the case); if the manuscript is about teaching methods, then a reference or control group is missing.

The introduction was partially rewritten to better reflect the chosen methodology. The study is a case study about teaching methods, not an experiment. The purpose of design-based research is to develop solutions to problems (in this case the challenges of data science education as described in the introduction).

(B) Present section "2.Methodology" is confusing: (b1) it is not clear to what research it refers; (b2) it needs references to the employed methods, namely the design-based research, and systemic action research methods.

See references [20-21].

(C) Results provided in Table 4 and Table 5 (i.e., students' feed-back) should undergo a qualitative analysis (https://en.wikipedia.org/wiki/Qualitative_research) instead of the [numbered] raw sentences' presentation. Moreover, some quantitative feed-back might be of interest.

The many limitations to the results of this study are listed in section 4.3. The study is merely a starting point. While I don't think it is flawed as such, it certainly does not pretend to be an outstanding example of the two methodologies (design-based research for the intent, and system action research for the data gathering) mentioned. I fully agree that a quantitative addition to the study would be of interest but I am unable to produce such an analysis with the available data.

(D) Part of discussions would better fit into the section of results. Additional comments: (a) I would kindly recommend a thorough language revision – scientific writing follows distinct rules and style, with little similarity to oral communication or colloquial writing. (b) Numbering of level 2 headings (style "heading2") in section "1.Introduction" should be double-checked. (c) First and third person are confusingly interchanged in the introduction. (d) Use of italics should be consistent throughout the manuscript. (e) Format of the references should be revised.

All of these were addressed in the revised version of the paper.

Comments on the Quality of English Language - I would kindly recommend a thorough language revision – scientific writing follows distinct rules and style.

I apologize for the many mistakes in the original manuscript. They should be fixed in the revised version.

Round 2

Reviewer 1 Report

The authors have satisfactorily addressed most of my concerns

Minor editing of English language required

Reviewer 3 Report

The manuscript has been substantially improved.

The Author chose to submit a rebuttal, rather than make changes according to the major issues I identified -- as he pointed out, the final decision is up to the Academic Editor.

Please do not forget to double-check the references (e.g., an author was overlooked in the current reference 21).

The English is fine.